# Emergence of MR-Linac in Radiation Oncology: Successes and Challenges of Riding on the MRgRT Bandwagon

**DOI:** 10.3390/jcm11175136

**Published:** 2022-08-31

**Authors:** Indra J. Das, Poonam Yadav, Bharat B. Mittal

**Affiliations:** Department of Radiation Oncology, Northwest Memorial Hospital, Northwestern University Feinberg School of Medicine, Chicago, IL 60611, USA

**Keywords:** radiation treatment, MR-Linac, imaging, MRgRT, adaptive therapy, advances, pitfalls, outcome

## Abstract

The special issue of *JCM* on “Advances of MRI in Radiation Oncology” provides a unique forum for scientific literature related to MR imaging in radiation oncology. This issue covered many aspects, such as MR technology, motion management, economics, soft-tissue–air interface issues, and disease sites such as the pancreas, spine, sarcoma, prostate, head and neck, and rectum from both camps—the Unity and MRIdian systems. This paper provides additional information on the success and challenges of the two systems. A challenging aspect of this technology is low throughput and the monumental task of education and training that hinders its use for the majority of therapy centers. Additionally, the cost of this technology is too high for most institutions, and hence widespread use is still limited. This article highlights some of the difficulties and how to resolve them.

## 1. Introduction

Malignant tumor or cancer cells can be killed by cytotoxic drugs, radiofrequency, hyperthermia, cryosurgery, and ionizing radiation if a sufficient dose is given. Modern cancer care management uses a multimodality treatment approach. Radiation treatment has been used with curative and palliative intent, and it is estimated that nearly 60% of the cancer patients will need radiation at some point during the management of the disease. The biggest problem with radiation treatment is that it is nondiscriminatory and also irradiates surrounding normal tissues (organs at risk—OAR), producing normal tissue toxicities, and in many cases limits the use of tumoricidal dosage. The goal of precise radiation therapy is to optimize the dose to the tumor and minimize the dose to the OARs. For over 125 years, radiation treatment has evolved from kilovoltage beams, megavoltage beams, and particle beams (electrons, neutrons, protons) with various planning techniques from two-dimensional (2D), three-dimensional conformal radiation treatment (3DCRT), intensity-modulated radiation therapy (IMRT), and volumetric modulated arc therapy (VMAT). Radiation beams can be targeted to tumor locations to deliver tumoricidal doses; however, the tumor locations may vary due to organ motion. This is where treatment imaging plays an important role with new technology, image-guided radiation therapy (IGRT) that has evolved from fiducial markers and cone-beam CT (CBCT), as developed by Jaffray et al. [1].

For the management of cancer treatment, the key is to deliver sufficient doses to eradicate tumor cells. This has been achieved by directing radiation beams to the tumor using multiple beams from different directions to deliver a higher dose. The problem is that the tumor is not a static structure; rather, it is dynamic and depends on surrounding body organs, such as thorax structures that move based on the breathing cycle and pelvic organs that move based on rectal and bladder filling. These problems can be solved to some extent by placing fiducial markers and using image-based radiation treatment, also known as gated treatment. Ultrasound imaging (BAT system) has also been tried [2]. Additionally, the IGRT and radio transponder system, Calypso [3,4], has been attempted with limited success.

Additional issues include our understanding of the location of the disease for contouring, which is dependent on the clinician’s training and knowledge, but more so on the imaging technology tools, some of which are suboptimal. Target volume contouring in each disease site has profound implications that have been documented extensively in the literature with a wide range of target variability [5,6,7,8,9,10,11,12,13,14,15]. Computed tomography (CT) has been an imaging option, as it provides a tool to calculate doses based on imaging data. However, CT images have their limitations in that they lack discrimination of soft tissue and profound metal artifacts, as shown in Figure 1 and Figure 2, respectively.

Figure 1 provides detailed information of soft tissue visualization from CT and magnetic resonance imaging (MRI), where data from MRI are better for visualization of the structures such as the stomach and bowel. With proper imaging techniques as described by Bitar et al. [16], even these can be further improved. Figure 2 provides an example of the artifact in CT with high density and high atomic number (Z) medium. As CT images are created based on the attenuation properties of tissues, they have difficulties with prosthetic devices, which are common in older and cancer patients. Artifacts are part of the CT imaging, and they occur due to the photon starvations (attenuation) of the photon beam passing through high-density and high-Z materials that produce significant artifacts, as shown in Figure 2.

The metal artifact provides significant difficulty for radiation treatment contouring and treatment planning. Firstly, the visualization of the structures is impaired and CT numbers are obscured, thus providing poor dose distribution in most treatment-planning systems, as shown by Yadav et al. [17]. To overcome these problems, multimodality imaging has been advocated using image fusion. Image fusion is inherently problematic as it depends on rigid or deformable registration, and quality assurance (QA) software for quantitative errors have not yet been developed [18,19,20,21].

MRI has been the backbone for diagnostic radiology due to the superior imaging of soft tissues (Figure 1); however, its role in radiation oncology has initially been limited, mainly due to the availability of scanners and reimbursement issues. Apart from the above considerations, more important issues related to MRI are dose calculations, as we cannot compute doses on MRI images due to the lack of electron density. Radiation oncology has relied on CT data for accurate dose calculation with ever-improving calculation algorithms, from Batho to Monte Carlo.

Radiation therapy dose calculations are dependent on the electron densities that are derived from CT data. There are no simple solutions for electron density generation from MR imaging. This is an ongoing project with no satisfactory solution. There has been significant progress in using MRI images to make them compatible with CT for dose calculation. These images are called pseudo or synthetic CT (sCT). The difficulties are that tissue characterization in MRI is dependent on the magnetic fields and imaging sequence, unlike the attenuation coefficient in CT images. An early approach to sCT started with bulk density approaches [22,23]. The Dixon approach, based on MRI parameters of soft tissue, fat, water, and air, has been relatively satisfactory [24,25,26] for soft tissues within ±3% accuracy in dose calculation. However, these approaches do not provide satisfactory results for organs containing bone. For head and neck cases, Korhonen et al. [27] used a polynomial fit approach to the MRI signal in and around the bone to create sCT. There are also two commercial sCT-generation products: Phillips (Philips Medical System, Amsterdam, Netherlands) and MRI Planner (Spectronic Medical AB, Helsingborg, Sweden), which have been successfully reported for soft tissues [28,29] within ±0.5% in general and up to ±1% overall dose difference accuracy, but data for high-density materials such as bone are still lacking. A lot of studies using deep learning and artificial intelligence (CNN, UNET, GNN, etc.) have been proposed and continue to be explored [28,30,31,32,33,34,35,36,37]. However, most scientists feel that additional research is still needed to make commercial systems for clinical use. Dose calculation on MRI images is becoming accurate, and it is expected that in the near future, MRI-only dose calculation could be as accurate as CT data, thus eliminating the need for CT scanning for many disease sites in radiation oncology.

To overcome image registration and provide MRI images for radiation treatment, a hybrid MR-Linac has been developed. There are several systems available, which have been described by Das et al. [38]. For clinical use, Elekta (Elekta, AB, Stockholm, Sweden) introduced the Unity system, with a 1.5 T magnetic field [39,40] with a 7 MV photon beam; and the ViewRay MRIdian (ViewRay, Mountain View, CA, USA) 0.35 T, originally introduced as a Co-60 based unit in 2014 but now replaced with a 6 MV linear accelerator [41]. Within a short period of time, there have been over 100 installed machines and close to 160 in preparation. During this time, scientific and clinical sites have been providing data either through respective consortium or personal research work. The growth in publications is truly exponential, as shown in Figure 3.

*Frontiers in Oncology* published a special issue in 2020 and *Journal of Clinical Medicine* (*JCM*) is publishing a special issue in 2022 devoted to MR-Linac. This special issue, “Advances of MRI in Radiation Oncology”, has already published over 15 papers that include various topics such as the economics of MR-Linac, dose escalation, interface dosimetry, and adaptive therapy, and includes site-specific information such as prostate, head and neck, soft tissue sarcoma, and rectum from both camps (Unity 1.5 T and MRIdian 0.35 T). This issue has also covered some important information on the radiobiology of rectal cancer. This scientific growth (Figure 3) indicates an amazing adaptation of this technology in radiation oncology. These hybrid systems in radiation oncology open up a new frontier in the exploration of new clinical science for radiation treatment. In this paper, we have highlighted a few important successes and challenges of MR-Linac.

## 2. Motion Management

Organ motion in radiation treatment is an important factor that should be considered; unfortunately, motion management is still in its infancy. MRI could offer unique opportunities that can provide images of motions using ultra-short echo imaging [42] or cine imaging. Inadequate dose delivery in radiation oncology is primarily due to the uncertainty of the tumor position, thus creating a larger margin for treatment. With MR-Linac, motion management is an integral part of the process where 2D cine images are acquired in real time. Structures (target and tracking structure) can be outlined and based on the inherent gating mechanism; treatment delivery is only possible when the target is in phase. One such example is shown in Figure 4a the beam is in pause condition when the outline structure is not aligned; and Figure 4b treatment is delivered when the structure is in phase. Such a capability is possible on the ViewRay MRIdian system using cine images where the beam trigger is within milliseconds without any dosimetric deviation, unlike in the old days where significant dose differences were noted in most machines [43,44].

## 3. Paradigm Shift

MR-Linac has created a paradigm shift in radiation oncology for managing soft tissue cancers. It integrates MRI images, on-line contouring on pretreatment MRI images, dose calculation on sCT derived from MRI, real-time planning, and gated therapy based on real-time cine’ images. It is expected to reduce collateral tissue damage by appropriate localization, reduction in margin, and on-line adaptation. MR-Linac may be a step towards improving the outcome by delivering accurate doses with focus radiation dose targeting. However, it adds complexity in real time for modulated therapy. Traditionally, volumetric modulated arc therapy (VMAT)/IMRT may require 1–2 weeks to plan, depending on the institutional priority for volume delineation, planning and QA, but in MR-Linac, it can be performed in near real time (30–50 min) depending on the efficiency of the staff. A new concept, MR-guided radiation therapy (MRgRT), has been introduced for real-time adaptive therapy where treatment is adapted based on MRI images [45,46,47,48,49]. However, this process is time-consuming and needs the accommodation of a suitable workflow, as described by Kerkmeijer et al. [50].

## 4. Success

Within a short period of time there have been a lot of publications on MR-Linac technology, developments, and innovation for patient treatment, as described in Figure 3. The biggest success of this technology is in visualizing tumors while treating, thus making it effective and accurate. Superior soft tissue contrast provides localization accuracy. It also eliminates fiducial marker placement, thus eliminating cost and burden. Imaging and motion management helps in most diseases, such as breast, head and neck, lung, liver, pancreas, rectum, prostate, and bladder cancer, to mention a few.

Even though clinical outcome data are missing due to the short follow up time, consortiums of both systems have embarked on clinical trials and are expected to produce positive outcome data. Randall et al. [45] quoted many outcome references. Hehakaya et al. [51] used an economical model and showed that MR-Linac is cost-effective with reduced complications. This is due to better targeting and eliminating of OAR from high-dose regions. Alongi et al. [52] reported improved quality of life with a 1.5 T system for prostate cancer. Additionally, Cuccia et al. [53] showed that MR-Linac hypofractionated treatment provides a satisfactory outcome with minimum toxicity.

Weykamp et al. [54], using the 0.3 T system for MR-Linac for hepatocellular carcinoma, showed that 5-fraction hypofractionation was well-tolerable with minimum toxicity with 2 years of overall survival of 82%, which is an impressive outcome. For oligometastatic cases arising from breast cancer, Tan et al. [55] showed better outcomes with minimal toxicity, as reported by the patients. The most difficult disease site, pancreas, with poor outcome, has been the main theme for MR-Linac, as it can provide better visualization, targeting, and adaptive therapy to minimize normal tissue toxicities. Chuong et al. [56] showed 2 years’ follow-up data in pancreatic cancer with an overall survival of 45.5% and with grade 3 toxicity of a mere 4.8%.

It is expected that over a short period of time, MR-Linac consortiums will provide clinical trial data favorable to this technology; however, some system drawbacks are discussed below that need to be overcome collectively.

## 5. Challenges

MR-Linac challenges are many, relating to imaging regions of low electron density such as the lungs, limited field of view and field size, time-intensive adaptive planning, a limited pool of standard immobilization devices, inability to deliver VMAT plans, and inability to treat patients with implants and metallic as well as cardiac devices. Currently, only one photon energy is available (6 MV for MRIdian and 7 MV for Unity); thus, one could argue that it might limit a larger pool of patients. However, it is known that for IMRT, as long as the number of beams is greater than seven, the beam energy is inconsequential, as supported by Pirzkall et al. [57], and only a low-energy beam may be sufficient. Another bottleneck could be the learning curve for different members of the team, including clinicians, physicists, dosimetrists and therapists involved. Some of the other challenges are described as below.

**Electron return effect (ERE):** Magnetic fields produce Lorentz’s force and are especially significant for high magnetic fields, providing significant doses to skin–air or lung–tissue interfaces. This has been addressed in literature [58,59,60,61]. However, if ERE is properly modeled in the treatment-planning system, it may not be of clinical concern. Additionally, to eliminate ERE, several precautions, e.g., bolus and large number of beams should be used. With these preventative precautions, skin toxicities due to ERE are not reported. As expected, the ERE is much lower in the 0.35 T MRIdian system. Nonetheless, users should be aware of the beam parameters affecting the skin dose [62].

**Low-field imaging:** It is a common belief that a high magnetic field is needed for an ideal signal-to-noise ratio for better image quality. However, the high-field-created ERE [58,63,64] may not be suitable in treatment due to dosimetric discontinuities, especially in lung–tissue interfaces. It is known that low-field imaging is superior for lung cases with respect to possible geometric distortions because of changes in magnetic susceptibility and chemical shift (susceptibility artefacts and chemical shift artefacts increase linearly with the magnetic field). Low-field imaging may provide dosimetric advantages, but needs additional proof that may be forthcoming in future literature. Further research and adaptability of low-field imaging is needed in MRL.

**MRI-based immobilization systems:** Immobilization plays an important role in radiation therapy treatment. Chandarana et al. [65] provided some information for MRI; however, due to eddy currents [66] in these devices, image quality suffers. Advances in immobilization are required that do not degrade image quality. Unfortunately, most devices produce MRI artifacts; hence, most patients are treated without immobilization, adding to patient discomfort. There is progress in this area, as an Austrian company, IT-V (www.it-v.net), is making immobilization devices that do not produce artifacts. The artifacts are also compounded by the MR-Linac bore sizes, and hence adding immobilization devices reduces the space for patients in terms of the MRI field of view.

**Autocontouring and planning:** Target and OAR contouring is an arduous and time-consuming process. It becomes even more difficult when on-table adaptive therapy is performed. The same can be said for treatment planning comprising optimization and dose calculation [67]. There is a critical need for autocontouring and planning in MR-Linac systems. Autocontouring may be possible in cine MRI images, as described by Fast et al. [68].

**Economics/reimbursement:** MR-Linac systems are currently several times more expensive (US$10–15 M) and require a substantial capital investment. To recoup the cost, either the number of treated patients must increase or reimbursement should become more favorable. Some of the economics of MRL and adaptive therapy have been addressed by Palm et al. [69].

**Throughput:** Most MR-Linacs are relatively slow processes and can treat only 6–8 patients in an 8 h/day. This is an impediment to the system, as on a standard machine one could treat 20–30 patients/day. On the other hand, MR-Linacs are being used primarily for hypofractionation (3–5 fractions), and reimbursement is better due to adaptive planning. It should also be emphasized that this is a special machine with a unique niche and cannot be compared to the regular machines.

**Dose rate/time:** Time is of the essence for the comfort and treatment of the patient and should be minimized. Currently available dose rates in MR-Linacs are relatively low (6 Gy/min). This needs to be optimized to patient on-couch time and throughput for radiation oncology centers.

**VMAT:** IMRT and VMAT are inverse planning processes in which IMRT is a discrete (step-shoot) beam angle and VMAT is a continuous arc therapy, which have been described in [67,70]. Current MR-Linac systems do not include VMAT capabilities due to inherent image distortion for real-time tracking. It is well-known that VMAT saves a significant amount of time and is recommended for modulated beams mainly due to its time-saving properties, lower monitor unit, and lower whole-body dose.

**MPR:** Unlike CT images, where digitally reconstructed radiographs (DRR) [71] and multiplanar reconstruction (MPR) [72] are common tools, but not available in MR-Linacs. MPR adds another flexibility of viewing the structures in any plane, which may be advantageous to spare OAR if available in MR-linac systems.

**High frame rate:** For imaging and treatment of organs in motion such as the heart, a high frame rate is needed [73]. Currently, the system is capable of either 8 FPS (MRIdian) or 5 FPS (Unity).

**Training for manpower:** As diagnostic and therapeutic radiology (radiation oncology) split in the 1980s, knowledge of imaging and treatment modalities is not freely shared. This impacts the clinical and technical components of the knowledge of practitioners and other experts. There is an urgency for additional manpower as well as cross-training for the greater success of MR-Linac [38,74]. Collaboration between vendors and users is needed to develop educational programs.

## 6. Conclusions

This special issue of *J Clin Med* provides a glimpse of the future to come in terms of the use of MR images for radiation treatment. MR-Linac is emerging as one of the important tools in radiation oncology. It provides soft tissue visualization during treatment, unlike CBCT. This allows on-table adaptive therapy, thus escalating the dose and possibly improving the outcome. Currently, the technology is in nascence. The full capabilities of MRI imaging using advance imaging sequences have not been utilized and the clinical outcome data are not mature. However, it is expected that this will reduce complications and increase overall survival.

## Figures and Tables

**Figure 1 jcm-11-05136-f001:**
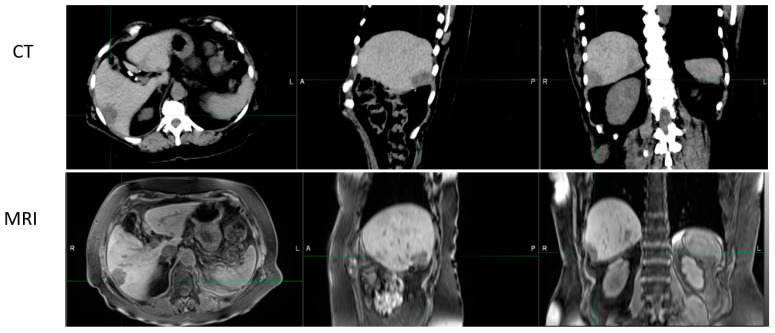
Axial, sagittal coronal images of abdomen region. Upper panel, CT data; lower panel, MRI image. Please note that soft tissue structures are more discernible in MRI than CT images.

**Figure 2 jcm-11-05136-f002:**
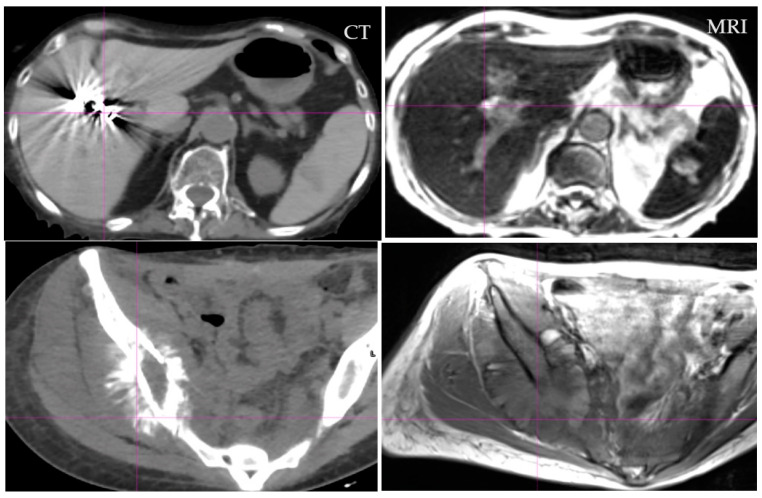
Two patients’ CT and MRI axial images. Note the significant artifact in CT image, which is nearly nonvisible in both 0.35 T (right top) and 1.5 T (left bottom) MRI image. Artifact is dependent on the density and atomic number (Z) of the medium.

**Figure 3 jcm-11-05136-f003:**
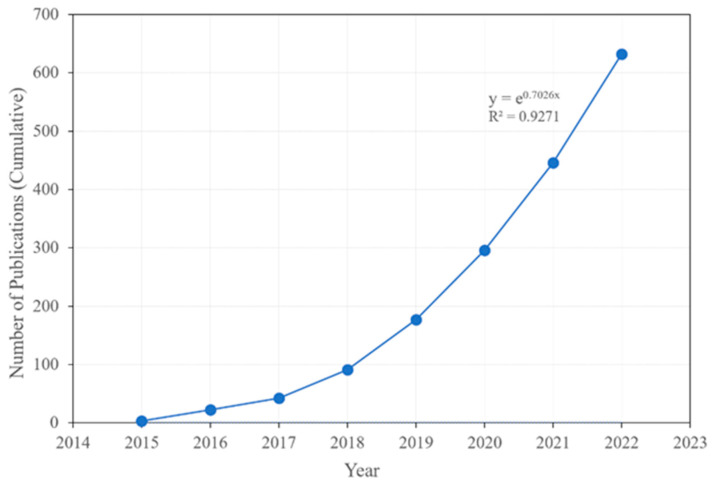
Growth of publications reported in PubMed since inception of MR-Linac indicating an exponential growth.

**Figure 4 jcm-11-05136-f004:**
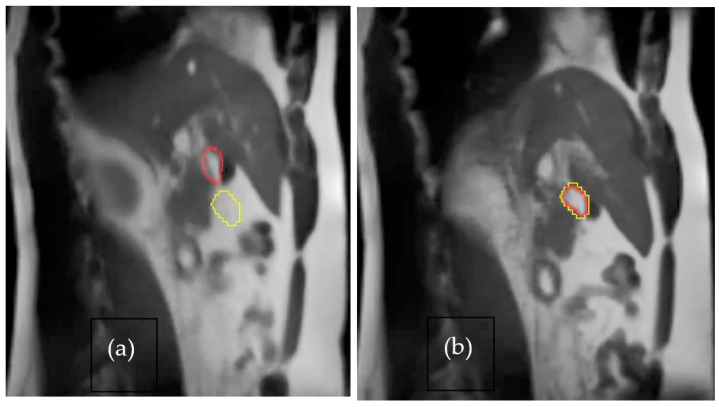
MRI cine image showing motion; (**a**) tracking structure (red) is out of defined boundary (yellow) and (**b**) with breath holding, tracking structure is in the boundary, which is automatically synchronized by the treatment delivery in MR-Linac from ViewRay.

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
