# Peer review of "Emergence of MR-Linac in Radiation Oncology: Successes and Challenges of Riding on the MRgRT Bandwagon"

_jcm, 2022, doi:10.3390/jcm11175136_

Round 1

Reviewer 1 Report

Thanks for that nice overview about the advantages and shortcomings of MR-Linacs used in radiation therapy. I would suggest some minor changes for further improving the quality of the paper:

Lines 92-97: How to generate the correct electron densities for dose calculation in a workflow that is mainly based on MR-imaging is a frequently discussed subject with no final answer. The standard methods at clinical MR-Linacs are deformable image registration between planning-CT and planning-MRI or the usage of bulk density assignments. For some regions of the body (when no lung is involved) also overwriting the whole patient outline with rED 1 leads to relatively accurate dose values (see e.g. J. Kim et al “Dosimetric evaluation of synthetic CT relative to bulk density assignment-based magnetic resonance-only approaches for prostate radiotherapy”, Radiat Oncol. 2015; 10: 239). But I think there is an issue that could be more important for correct dose calculation and should be mentioned in the article: the accurate definition of the patient outline. Due to limited field of view and distortions possible at the image borders, the patient outline has to be checked carefully. Deviations from the real patient surface could have more impact on the correct dose than inaccurate density assignments.

Lines 149-150: I strongly disagree that a VMAT/IMRT “traditionally” requires 2 weeks to plan. I think 1 week is a good time frame but in clinical reality often 2 days have to be sufficient. And there are also cases where the planning CT scan is done in the morning and first treatment in the evening of the same day. For the initial treatment plan at a MR-Linac the same time is needed for planning than for a conventional VMAT/IMRT plan (about 1 week). But the online adaptive planning process is (and has to be because the patient is still lying on the treatment couch) much faster because you start with an already existing plan for a (hopefully) similar patient geometry. The time scale here is about 20 to 40 minutes I guess.

Line 189: As you use normally a large number of incident beams with different gantry angels I could not support the statement that the photon energy (6 MV or 7 MV at MRIdianLinac or Unity, respectively) confines the pool of patients.

Lines 192-195: It should be also mentioned that ERE is calculated correctly by the TPS for MR-Linacs and therefore possible adverse dose distributions could be detected. On the other hand if many beam angels are used even at 1.5 T the effect of ERE is rarely pronounced.

Lines 196-200: Low field imaging could be superior for lung imaging and in respect to possible geometric distortions because changes in magnetic susceptibility and chemical shift (susceptibility artefacts and chemical shift artefacts increase linearly with B0). With respect to SNR and imaging frequency a higher B0 shows advantages. Dosimetric effects: see above!

Lines 202-207: the Austrian company IT-V (www.it-v.net) sells most of their standard immobilization equipment (mainly used for Elekta linacs) in a MR-compatible version for Unity as well as for MRIdianLinac. Artefacts are not the problem with these devices. The real problem with immobilization devices is the very limited bore size at MR-Linacs!

Lines 221-223: It is completely true that only about 8 patients can be treated in an 8 hour shift at a MR-Linac. But for comparing throughput to a conventional linac it has to be taken into account that MR-Linac patients are normally treated with a hypofractionated scheme. Typically 3 to 8 fractions are used in contrast to about 20 to 30 fractions at a conventional linac. For that reason the patient number integrated over a longer period is not so much smaller for a MR-Linac than for a normal linac!

Lines 237-239: Why you would like to have MPRs at a MR-Linac?

Lines 244: It also should be mentioned that not only the training for the manpower is a crucial aspect but also the needed manpower itself. Especially for the online adaptive process a radiation oncologist and a dosimetrist/medical physicist (country dependent) should/has to be in place at the MR-Linac for every fraction.

Author Response

Response in in enclosed file.

Reviewer 2 Report

I would like to thank you the Authors for the interesting commentary on a worthy topic

The Authors well summarize the main issues that the present special issue is expecting to analyze

I have no major comment on the text, some minor comments are following:

Legend figure 1 and through the text: please be consistent correct MR to MRI

Page 3 line 81 magnetic resonance imaging (MRI) move to the first time MRI is written, page 2  line 58.

Page 7 line 248; collabration correct to collaboration

Author Response

Response is in enclosed file combined with referee 1
